

# Restoring lepidopteran diversity in a tropical dry forest: relative importance of restoration treatment, tree identity and predator pressure

Lizet Solis-Gabriel[1], Wendy Mendoza-Arroyo[1], Karina Boege[2],* and Ek del-Val[1,3],*

[1] Instituto de Investigaciones en Ecosistemas y Sustentabilidad, Universidad Nacional Autónoma de México, Morelia, Michoacán, Mexico
[2] Instituto de Ecología, Universidad Nacional Autónoma de México, Ciudad de México, Ciudad Universitaria, Mexico
[3] Escuela Nacional de Estudios Superiores Unidad Morelia, Universidad Nacional Autónoma de México, Morelia, Michoacán, Mexico
* These authors contributed equally to this work.

Corresponding author
Ek del-Val, ekdelval@iies.unam.mx

## ABSTRACT

Tropical dry forests (TDFs) have been widely transformed by human activities worldwide and the ecosystem services they provide are diminishing. There has been an urgent call for conservation and restoration of the degraded lands previously occupied by TDFs. Restoration experiences aim to recover species diversity and ecological functions. Different restoration strategies have been used to maximize plant performance including weeding, planting or using artificial mulching. In this investigation, we evaluated whether different restoration practices influence animal arrival and the reestablishment of biotic interactions. We particularly evaluated lepidopteran larvae diversity and caterpillar predation on plants established under different restoration treatments (mulching, weeding and control) in the Pacific West Coast of México. This study corroborated the importance of plant host identity for lepidopteran presence in a particular area. Lepidopteran diversity and herbivory rates were not affected by the restoration treatment but they were related to tree species. In contrast, caterpillar predation marks were affected by restoration treatment, with a greater number of predation marks in control plots, while caterpillar predation marks among plant species were not significantly different. This study highlights the importance of considering the introduction of high plant species diversity when planning TDF restoration to maximize lepidopteran diversity and ecosystem functioning.

## INTRODUCTION

Ecological restoration aims to recover species diversity and ecological functions (*Society for Ecological Restoration, 2007*; *Howe & Martínez-Garza, 2014*). Different restoration strategies have been used to maximize plant survival and performance including

weeding, planting or using artificial mulching. In general, treatments that enhance soil water content and minimize competition with background vegetation are the ones showing better results for plant performance (*Chalker-Scott, 2007*; *Barajas-Guzmán & Barradas, 2011*). However, when considering other aspects for restoring ecological functions such as the arrival of primary and secondary consumers, very few investigations have evaluated different restoration treatments. The recovery of animal populations is fundamental for restoring ecological functions (*Noreika et al., 2015*; *Jones & Davidson, 2016*). Therefore, there is a need to understand if differences in vegetation performance due to different restoration treatments are translated into animal communities and further into ecological functioning.

Lepidopterans are an important group of invertebrates in tropical forests because they are a very diverse group and function as herbivores when larvae and pollinators as adults. As herbivores they consume significant quantities of leaf tissue (*Novotny et al., 2002*, *2004*, *2006*; *Dyer et al., 2007*) and as moths and butterflies account for the pollination of at least 10% of plant species in tropical dry forests (TDFs) (*Haber & Frankie, 1989*). Therefore, when considering the restoration of ecosystems, lepidopterans are a group that should be considered since they can help safeguard plant reproduction. Also, lepidopterans represent a significant food source for predators in these forests and so are needed to restore the insectivore community.

Tropical dry forests are one of the most important vegetation types in Latin America. They used to cover 50% of land (*Murphy & Lugo, 1986*; *Sánchez-Azofeifa et al., 2005*). In Mexico in particular they covered 37% of the country, however due to anthropogenic activities such as agriculture and cattle farming (*Trejo & Dirzo, 2000*), only 30% of the original area remains pristine. The current scenario involves a mosaic of large areas of degraded lands surrounded by secondary forests and few federal and state preserves (*Sánchez-Azofeifa et al., 2009*). Therefore, there is an urgent need to restore degraded lands to conserve ecological functions and guarantee ecosystem services (*Ceccon et al., 2015*). Controversies around restoring TDFs have arisen due to their relatively high successional speed. Some authors argue that fencing against cattle should be sufficient to ensure forest recovery while others advocate for active interventions involving planting of native tree species (*Aide et al., 2000*; *Gonzáez-Iturbe, Olmsted & Tun-Dzu, 2002*; *Burgos & Maass, 2004*; *Lebrija-Trejos et al., 2010*). Recent investigations have found that fencing against cattle in a TDF of Southern Mexico was more important for lepidopteran recovery than planting (*Juan-Baeza, Martínez-Garza & del-Val, 2015*). However, active planting has been shown to speed up plant regeneration and lepidopteran arrival in other restoration experiences (*Hernández et al., 2014*).

With the aim of understanding the relative contributions of different restoration treatments for biodiversity and ecological function recovery, in this paper we investigated whether different restoration treatments in the TDF: (1) have differential impacts on lepidopteran communities associated with introduced plants; (2) lead to differences in herbivory rates; and (3) have differential impacts on predation rates on model lepidopteran larvae.

## METHODOLOGY

Our experimental area is situated in the central Pacific coast of Mexico in the surroundings of the Chamela-Cuixmala Biosphere Reserve (CCBR, 19°29′N, 104°58′ 105°04′W), Jalisco, Western Mexico, in pastures formerly covered with TDFs in the La Huerta municipality. The main vegetation in the area is TDF with a canopy height between 5 and 10 m, and semi-deciduous forests along riparian zones; dominant plant families are Leguminosae, Euphorbiaceae and Rubiaceae (*Lott, Bullock & Solís-Magallanes, 1987*; *Noguera et al., 2002*). Mean annual temperature is 24.6 °C (1978–2000) with a monthly oscillation of 4.3 °C, and mean annual precipitation of 731 mm (*García-Oliva, Camou & Maass, 2002*). The rainy season is concentrated from July to November (*Noguera et al., 2002*) followed by an intense dry season where precipitation is almost 0 mm. The soil types are eutric regosols, which are highly drained, causing poor water retention (*Noguera et al., 2002*). The surrounding area of the reserve consists of a mosaic of secondary succession forests, agricultural fields and cattle pastures (*Sánchez-Azofeifa et al., 2009*). The TDF found at Chamela-Cuixmala is considered one of the most diverse of its kind with 1,200 plant species, comprising a high percentage of endemism (*Lott, Bullock & Solís-Magallanes, 1987*; *Trejo & Dirzo, 2000*). The invertebrate inventory is quite small; however, 1,877 invertebrate species have been described, 583 of which are lepidopteran species (*Pescador-Rubio, Rodríguez-Palafox & Noguera, 2002*).

The restoration area where this investigation took place is located on private land that had been used as cattle pasture for ca. 50 years, but since 2010 the land was put aside for ecological restoration. Ten hectares covered with exotic pastures were restored using 11 native tree species following a blocked experimental design that included three restoration treatments: plastic mulching, weed removal and a control group. Planted species were *Cordia alliodora* (Ruiz & Pav.) Oken, *Cordia elaeagnoides* D.C., *Caesalpinia eriostachys* Benth., *Caesalpinia platyloba* S. Watson., *Caesalpinia pulcherrima*, *Lysiloma microphylla* Benth., *Apoplanesia paniculata* C. Presl, *Leucaena leucocephala* (Lam.) de Wit, *Guazuma ulmifolia* Lam., *Gliricidia sepium* (Jacq.) Kunth ex Walp. and *Heliocarpus pallidus* Rose. The treatments were replicated five times in each of five sites with a distance no greater than 1 km ($N = 25$ plots; see *Saucedo-Morquecho, 2016* for experimental details, Fig. 1). Ten individuals approximately 1 m tall of each species were planted in a 30 × 36 m plots in a 3 × 3 grid ($N = 30$ individuals/species/site/treatment, a total of 4,950 plants). To facilitate mycorrhizal colonization, at the time of planting we added ca. 300 g of soil collected at the sites where maternal trees were established. Plots were randomly assigned to one of the following treatments: (1) Plastic mulching, which consisted of covering the soil with an agricultural use plastic before planting; (2) cutting grasses, which consisted of manually removing the vegetation around each sapling every three months during one year; and (3) no management after planting.

### Lepidopteran sample

In order to assess Lepidoptera larval diversity in the restoration treatments, in 2014, three years after the experimental set up and when plants were 2 m in height on average (*Saucedo-Morquecho, 2016*), we sampled a subset of the plots under the three legacy

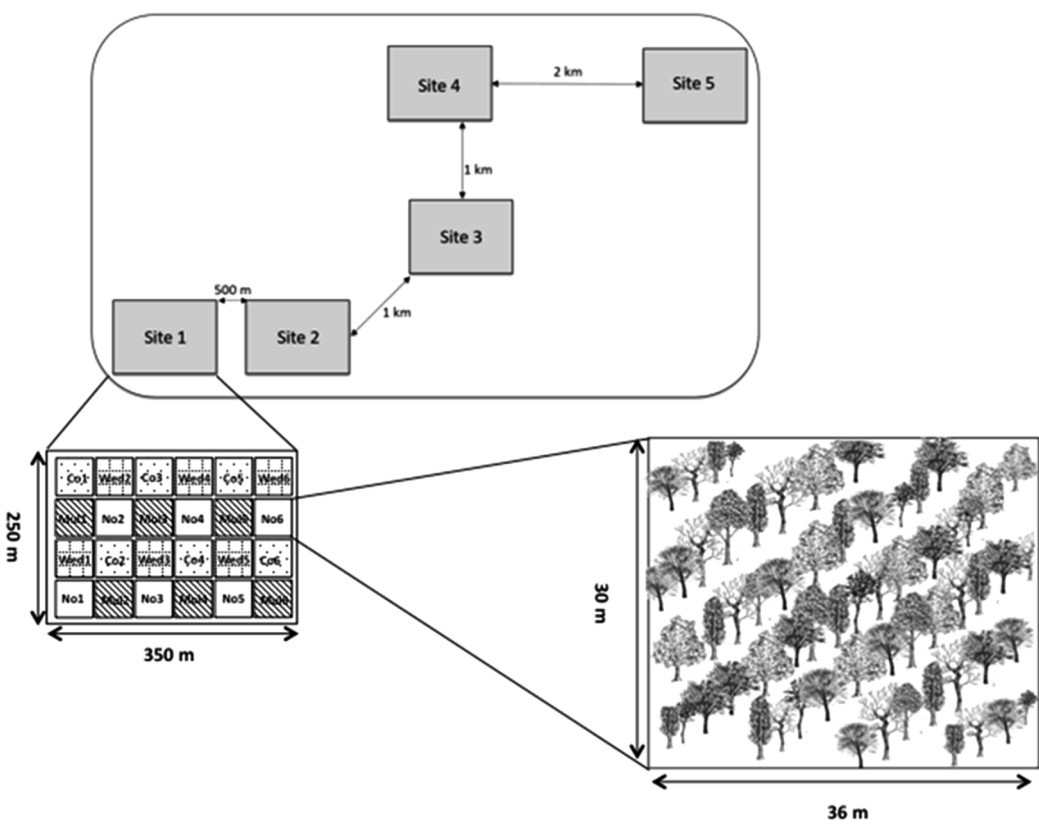

**Figure 1 Experimental set-up showing different experimental sites and restoration plots inside each site.** Restoration code treatments are as follows: Co, control with planting; Mu, planting with plastic mulching; Wed, planting with weeding; No, control with natural regeneration. Note that in this article we only report results from the Co, Mu, and Wed restoration treatments.

experimental treatments in three sites ($N = 3$ plots/treatment), including 11 experimental plant species. Sampling was conducted on four plants per tree species ($N = 44$) per plot ($N = 396$ plants). During the rainy season of 2014 (July–November) monthly censuses were conducted looking for lepidopteran larvae in all selected plants, searching for caterpillars on all leaves and stems. The presence of caterpillars were recorded, and if unknown they were collected, transferred to the lab and reared into adulthood to further identify the species.

## Herbivory rates

At the end of the rainy season (November), we estimated leaf area consumed by herbivores in five randomly selected mature leaves collected from the same plants used for herbivore censuses. However, only seven species could be assessed as the other four species (*Leucaena leucocephala, Caesalpinia pulcherrima, Caesalpinia eriostachys* and *Lysiloma microphylla*) have very small leaflets in which leaf damage is difficult to assess. Leaves were shade dried and scanned in the laboratory. Leaf area loss was assessed using the program SigmaScan Pro, then we calculated leaf area lost per plant, per species in the different restoration treatments.

## Caterpillar predation

During the rainy season of 2015, we evaluated lepidopteran larvae predation at the same restored sites. In this case, due to time constrains, we used five plants of nine species in only two of the legacy restoration treatments (plastic mulching and control) in two sites (we excluded *Cordia alliodora* and *Leucaena leucocephala* because of high mortality during 2015; total sampled plants = 180). To infer lepidopteran predation we used artificial clay caterpillar models as proposed by *Richards & Coley (2007)*. Caterpillar models were 3 cm by 0.5 cm thick. We used models in bright green and brown-yellow that mimic the most common caterpillar colors in the region. We decided to use two caterpillar colors since it has been reported that coloration plays an important role in predator behavior and we wanted to test this hypothesis for the TDF. For each experimental plant, we exposed four artificial clay caterpillars (two green and two brown-yellow), a total of 180 caterpillars per restoration treatment per site. Artificial caterpillars were fixed to leaf petioles or abaxial part of leaves using white glue. We exposed caterpillar models to predators for 24 h and then we estimated predation by evaluating marks on the clay models. Caterpillars with predation marks were photographed to be analyzed in more detail using a computer. Predation types were assigned following *Tvardikova & Novotny (2012)* proposal. We repeated the predation experiments four times between July and October 2015, once every month. Missing caterpillars were not included in the analyses since we do not know their final destiny; they accounted for 15% of clay caterpillar models. Missing caterpillars may have fallen from the trees because the glue was not strong enough or predators may have taken them away.

## Statistical analysis

Lepidopteran richness and abundance was analyzed using nested ANOVAs, with plant species and restoration treatment as the explanatory variables nested by plot/restoration treatment. To analyze lepidopteran community similarities between plant species and restoration treatments we obtained Bray–Curtis indices per plant species and per treatment, we then plotted the resulting dendrograms showing Bray–Curtis distances and performed a Mantel test with 100 permutations using the "vegan" library to assess the tree significance. Herbivory rates were analyzed using the percent leaf area damaged per plant transformed with arcsin. We also used a nested ANOVA using species and restoration treatment as the explanatory variables nested by plot/restoration treatment. We also performed a Pearson correlation between herbivory per plant and total lepidopteran abundance. Caterpillar predation was analyzed using a lme (linear mixed effect model) with total percent predation; green caterpillar percent predation and brown caterpillar percent predation as response variables and restoration treatment, with tree species and sampling month as explanatory variables. To analyze differences in predator type we used ANOVA with percent caterpillar predated as a response variable and presumed predator type, caterpillar color, restoration treatment, sampling month and their interactions as explanatory variables. All analyses were performed with R program version 2.14.0 (R-Core development).

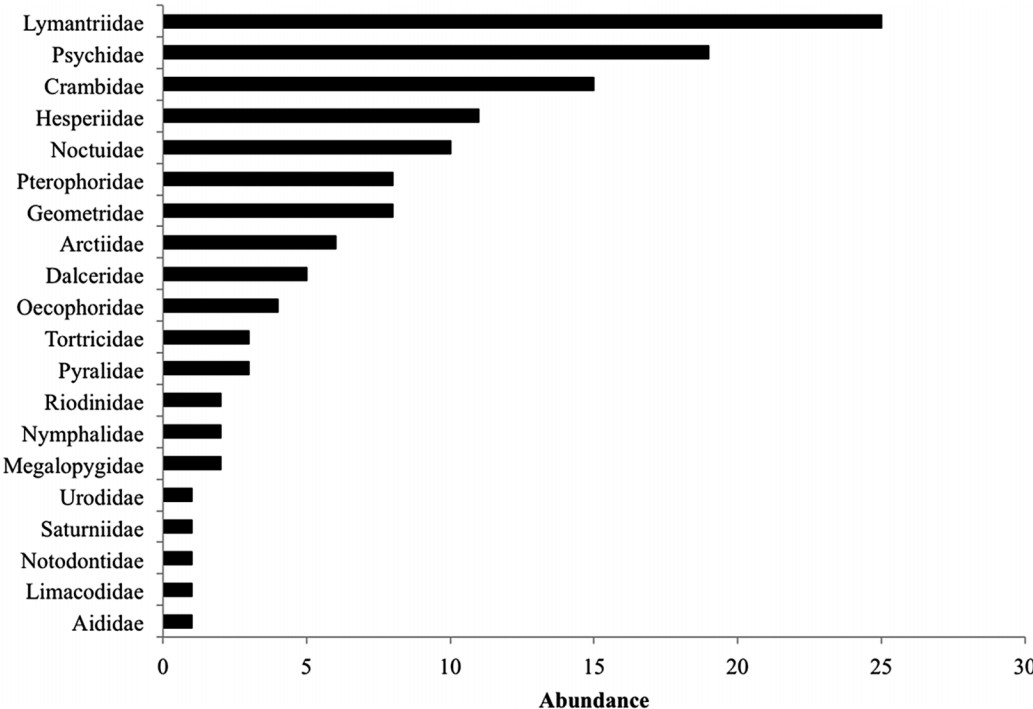

**Figure 2** Total caterpillar abundance per family including all restoration treatments.

**Table 1** Number of unique (diagonal in bold) and shared (below the diagonal) lepidopteran species between restoration treatments.

|          | Weeding | Mulching | Control |
|----------|---------|----------|---------|
| Weeding  | **14**  |          |         |
| Mulching | 14      | **10**   |         |
| Control  | 19      | 10       | **11**  |

## RESULTS

### Lepidopteran diversity

During the 2014 rainy season, we found a total of 234 lepidopteran larvae from 89 species (16 identified to species level, four to genus and 41 identified to family), 18 species comprised most individuals (44.8%). Lymantriidae, Psychidae and Crambidae were the best-represented families with 25, 19 and 15 individuals (Fig. 2). Lepidopteran abundance and richness are not affected by the restoration treatment ($F_{(2,42)} = 1.22$, $P = 0.3$) but were related to the particular tree species sampled ($F_{(10,42)} = 2.6$, $P = 0.01$), regardless of the restoration treatment (plant species vs. restoration treatment interaction: $F_{(18,42)} = 0.67$, $P = 0.81$). Many lepidopteran species (45%) were present only in one restoration treatment (Table 1) and the Bray–Curtis dissimilarity index between restoration treatments was also high ranging from 71% (control vs. weeding treatment) to 85% (control vs. mulching treatment). The Bray–Curtis dissimilarity index between plant species was very high ranging from 63% to 100%, suggesting that lepidopteran community composition was influenced by host identity (Table 2). Interestingly, lepidopteran communities

**Table 2 Bray–Curtis dissimilarity index between lepidopteran species associated to different plant species.**

| | Cordia eleagnoides | Cordia alliodora | Gliricidia sepium | Caesalpinia platyloba | Guazuma ulmifolia | Leucaena leucocephala | Caesalpinia eriostachys | Apoplanesia paniculata | Heliocarpus pallidus | Caesalpinia pulcherrima |
|---|---|---|---|---|---|---|---|---|---|---|
| Cordia alliodora | 0.93939 | | | | | | | | | |
| Gliricidia sepium | 0.90804 | 0.97058 | | | | | | | | |
| Caesalpinia platyloba | 0.91836 | 0.93333 | 0.92857 | | | | | | | |
| Guazuma ulmifolia | 0.87301 | 0.95454 | 0.69387 | 0.700000 | | | | | | |
| Leucaena leucocephala | 0.958333 | 1.000000 | 0.83132 | 0.955555 | 0.7966 | | | | | |
| Caesalpinia eriostachys | 0.937500 | 1.000000 | 0.82089 | 0.931034 | 0.8139 | 0.6428 | | | | |
| A. paniculata | 0.894736 | 0.894736 | 0.78082 | 0.942857 | 0.6326 | 0.7058 | 0.555556 | | | |
| Heliocarpus pallidus | 0.869565 | 1.000000 | 0.92592 | 0.860465 | 0.8596 | 0.95238 | 1.000000 | 0.9375 | | |
| Caesalpinia pulcherrima | 0.941176 | 0.866667 | 0.97101 | 0.870967 | 0.8666 | 0.73333 | 1.000000 | 0.9000 | 1.0000 | |
| L. microphyllum | 0.942857 | 0.875000 | 0.97143 | 0.87500 | 0.8695 | 0.93548 | 1.00000 | 0.90476 | 1.0000 | 0.7647 |

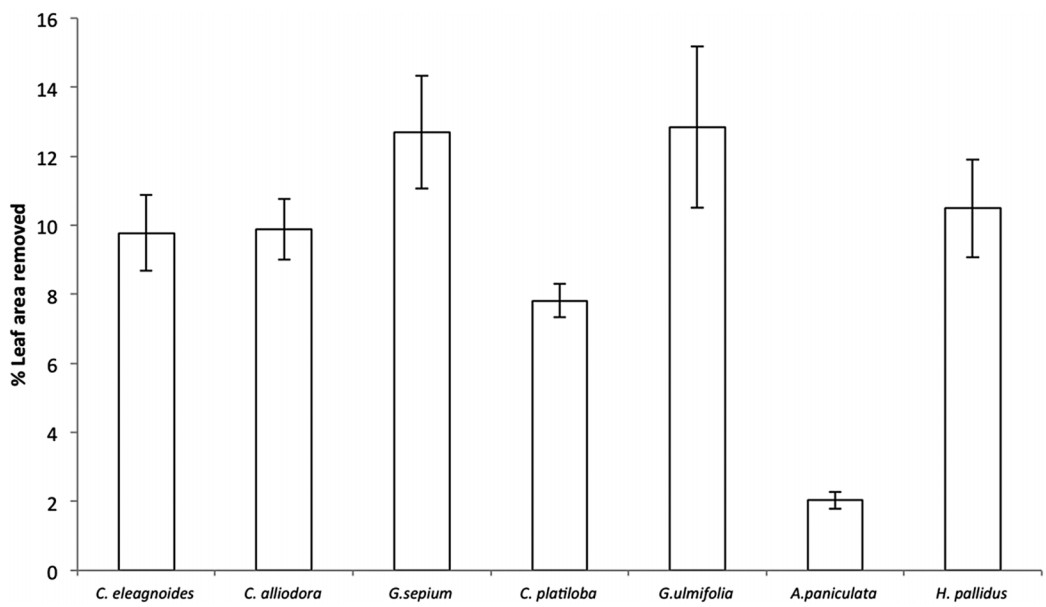

**Figure 3 Percent leaf area removed (mean ± EE) in trees, averaged across all restoration treatments in the TDF, $F_{(6,36)}$ = 22.7, $P$ < 0.001.**

associated with the different species of the same genus, such as *Caesalpinia* or *Cordia* did not form uniform groups; each *Caesalpinia* and *Cordia* species was found in a different branch of the cluster (Supplemental Information 1).

## Leaf damage by herbivores

Percent leaf area removed per species ranged from 2% in *A. paniculata* to 12% in *Guazuma ulmifolia* and *Gliricidia sepium* ($F_{(6,36)}$ = 22.7, $P$ < 0.001; Fig. 3). However, herbivore damage was not different as a function of restoration treatments ($F_{(2,4)}$ = 2.49, $P$ = 0.19) for any host species (plant species vs. restoration treatment interaction: $F_{(12,36)}$ = 1.03, $P$ = 0.45). Leaf damage did not correlate with caterpillar abundance per plant ($r$ = 0.003, $P$ = 0.96) or total caterpillar abundance per plant species ($r$ = 0.56, $P$ = 0.14).

## Caterpillar predation

During the 2015 rainy season, a total of 2,376 caterpillar clay models were exposed to predation in the restoration plots, of which 352 (14.8%) presented marks suggesting some type of predation (see Supplemental Information 2 for examples) and 359 (15.1%) disappeared during the experiment. Caterpillar predation on different host species ranged from 16% (in *Cordia elaeagnoides*) to 9% (in *Gliricidia sepium*) but because the variance was high in all species we found no statistical differences across plant species ($F_{(8,39)}$ = 0.656, $P$ = 0.72; Fig. 4). Caterpillar predation was greater on trees growing in the control treatment (56% vs. 44%; $F_{(1,132)}$ = 3.95, $P$ = 0.048). Also, predation during the rainy season was different between months, in both the control and mulching treatment plots the percentage of predated caterpillars was lower in July (3% and 2%, respectively)

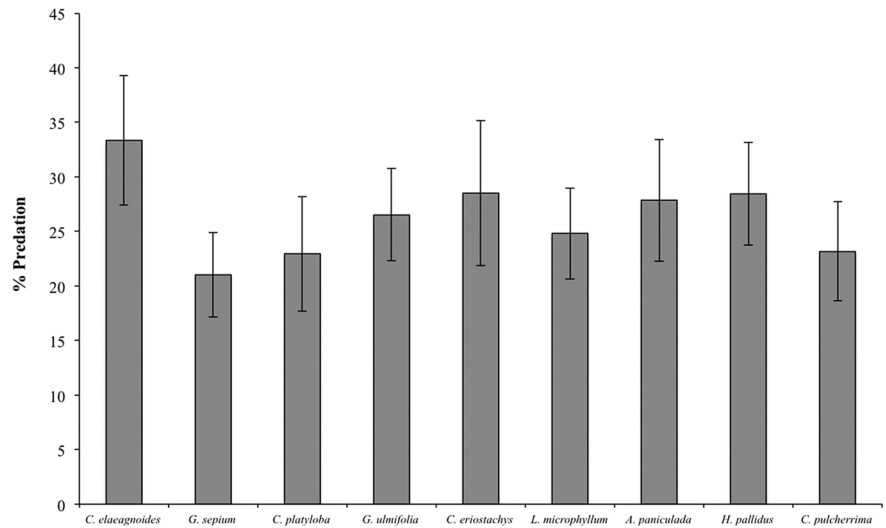

**Figure 4 Percent predation on clay models (mean ± SE) associated with different plant species during the rainy season of 2015, $F_{(8,39)} = 0.656$, $P = 0.72$.**

than later during the rainy season (13% in October for control plots and 10% in August for mulching treatment plots; $F_{(1,132)} = 4.71$, $P = 0.03$; Fig. 5A).

We found that color of artificial caterpillar clay models affected predation rates, where green models were more predated (56%) than brown-yellow ones (44%; $F_{(1,132)} = 7.31$, $P = 0.007$), irrespective of host species ($F_{(8,132)} = 0.129$, $P = 0.997$) or restoration treatments ($F_{(1,132)} = 0.277$, $P = 0.59$; Fig. 5B).

When analyzing the predator type attacking caterpillar clay models, we distinguished two general predation marks: (1) beak marks imposed by birds; and (2) marks imposed by invertebrates characterized by small holes or small scrapes presumably made by mandibles (Supplemental Information 2). Marks attributed to invertebrate predation were significantly more frequent than marks caused by bird predation (9.7% vs. 5% of predated caterpillars, respectively; $F_{(1,6)} = 40.41$, $P = 0.0007$). Temporal trends of bird and invertebrate predation marks showed different patterns, with bird predation being more important in August, while invertebrate peak predation was in September (month vs. predator type interaction: $F_{(3,6)} = 10.059$, $P = 0.009$; Fig. 6). Both types of predators marks were greater on green caterpillars but the difference between colors was more pronounced for bird predation marks (predator type vs. caterpillar color interaction: $F_{(1,6)} = 8.69$, $P = 0.025$).

## DISCUSSION

This study corroborated the importance of plant host identity for the recovery of caterpillar populations in restoration efforts. Even though plant species showed differences in performance depending on the restoration treatment applied in the site (*Saucedo-Morquecho, 2016*), lepidopteran species were not responsive to restoration treatments, but showed large differences among host plant species in terms of richness and abundance. This finding is similar to other studies that have found that lepidopteran

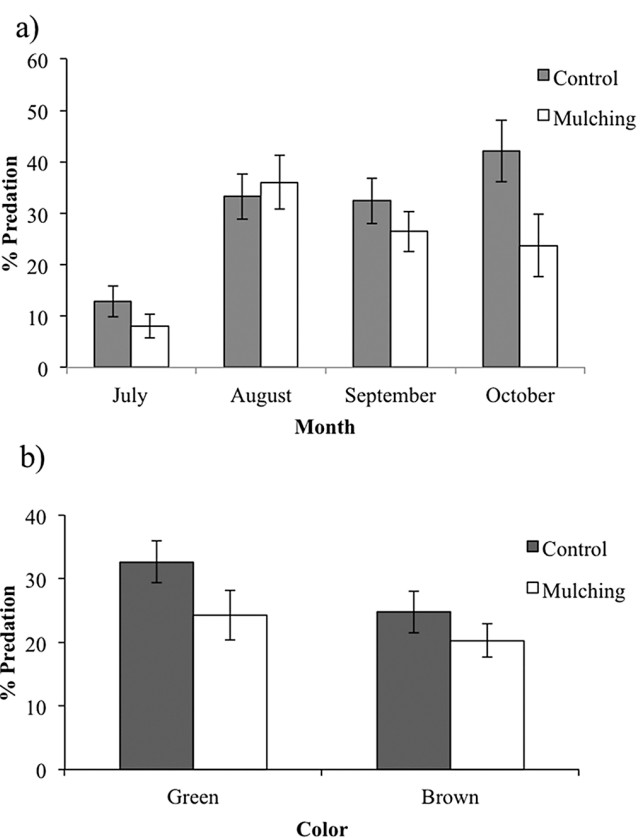

Figure 5 Percent predation on caterpillar clay models (mean ± SE) during the rainy season of 2015: (A) in the control and mulching restoration sites, month vs. predator type interaction: $F_{(3,6)}$ = 10.059, $P$ = 0.009, (B) on green and brown caterpillar clay models (mean ± SE) in control and mulching restoration treatments.

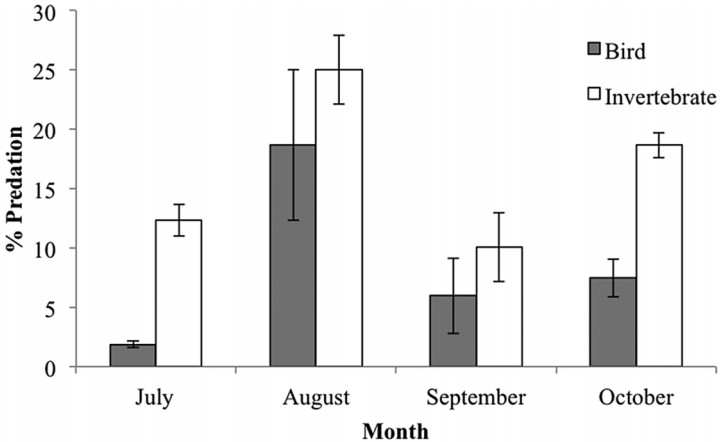

Figure 6 Percent predation caterpillar clay models (mean ± SD) per guild type (bird or invertebrate) during the rainy season of 2015.

communities are strongly determined by host species identity irrespective of land use history (*Hernández et al., 2014*; *Juan-Baeza, Martínez-Garza & del-Val, 2015*). Lepidopteran association with particular plant species is dictated by plant nutritional

quality, plant appearance and by predation experienced in particular plants. In particular, plant nutritional quality has been associated with nutrient concentration, secondary metabolites and physical defenses (thickness, trichomes and waxes) (*Dyer et al., 2007*). These characteristics are known to vary not only among species but also across sites (*Pennings, Siska & Bertness, 2001*; *Boege & Dirzo, 2004*). However, because we did not assess these traits, further studies are needed to test how they may have influenced the herbivore communities.

Leaf area removed by herbivores followed the same pattern observed in lepidopteran diversity; some tree species had greater damage than others (in particular *Gliricidia sepium* showed the highest percent of leaf area consumed). Since we did not find a significant correlation between lepidopteran abundance and leaf damage in individual plants or at the species level, it is possible that the observed damage can be due to other herbivores such as coleopteran larvae, grasshoppers or ants, which are known to be important herbivores in the Chamela TDF; this hypothesis also warrants further investigation. Interestingly, herbivore damage levels found in this study are similar to previous investigations in the region concentrated in conserved forests (*Dirzo & Boege, 2008*). Hence, we conclude that our restoration plots attracted herbivores with similar ecological functions to those found in mature forests (i.e., similar pressures on plants due to leaf consumption), and herbivores are not increasing their abundance in a disproportionate way behaving as pests. This finding is particularly relevant, since it has been suggested that restoration efforts may concentrate resources for herbivores and plants can fail to establish because of increased herbivore pressure (*King & Keeland, 1999*; *Blanco-García & Lindig-Cisneros, 2005*; *Sweeney, Czapka & Petrow, 2007*). In our case, leaf damage was not exacerbated and plants were not particularly affected, therefore the restoration efforts were not hampered by herbivores.

Our results showed a very high lepidopteran species turnover between restoration treatments and also between plant species. This result mirrors the lepidopteran beta diversity characteristic of Mexican TDFs (*López-Carretero, 2010*, *2014*). Due to this high diversity, TDFs represent a challenge for ecological conservation and restoration, hence we recommend ensuring high plant diversity and heterogeneity in lepidopteran conservation/restoration programs.

## Lepidopteran predation

Caterpillar clay models were useful to measure lepidopteran predation by birds and invertebrates in the restoration experiments. We were able to infer that invertebrate predation was stronger than bird predation for caterpillars irrespective of color or plant species. *Richards & Coley (2007)* and *Suzuki & Sakurai (2015)* with the same methodology also reported that invertebrates are the main predators in a tropical rainforest in Costa Rica and in Japan, respectively. However, *Sam, Koane & Novotny (2015)* in Papua New Guinea showed that the predator guild changed across an altitudinal gradient, where birds were more important at high altitudes and ants were more important at low altitudes.

We predicted that caterpillar predation should differ among host species due to differences in canopy cover, height and structure. Other studies investigating insectivorous bird visitation rates to tree species have found that they prefer certain species, in particular the ones with greater insect abundance (*Gantz et al., 2015*) or with higher canopies (*Fink et al., 2009*). However, it is likely that young saplings planted at the same time in our experiment did not have pronounced architectural differences yet and this may have obscured possible predator preferences. Further investigation is needed to understand the relative importance of predation for herbivory at a plant community level, since we measured herbivory and predation rates in different years.

## CONCLUSION

This study concurs with previous restoration experiences in that restoring TDF is a viable option to recover biodiversity and highlights the importance of including a diverse community of plants to enhance biodiversity recovery. Although restoration treatments did influence plant growth (*Saucedo-Morquecho, 2016*), they did not scale-up to influence lepidopteran communities and predation rates. Hence, the reestablishment of ecological functions was independent of initial restoration treatment. It appears that once plants are established, if the restoration outcome is close to a conserved forest, as is the case in this study, herbivores and predators are able to colonize and resume biotic interactions. In this context, we suggest the use of the most economical option for future restoration efforts.

## ACKNOWLEDGEMENTS

We are grateful to Manuel Valdes, Lenin Machuca, Nallely Luviano, Juan Pablo Martínez and Gustavo Verduzco for their expert field assistance. The authors gratefully acknowledge logistical support from Wolfgang Hahn, Ari Nieto, and the staff of the Chamela Biological Station (UNAM).

### Funding

This work was supported by PAPIIT-UNAM program (No. IN205013) to Ek del-Val. The funders had no role in study design, data collection and analysis, decision to publish, or preparation of the manuscript.

### Grant Disclosures

The following grant information was disclosed by the authors:
PAPIIT-UNAM program: IN205013.

### Competing Interests

The authors declare that they have no competing interests.

## Author Contributions

- Lizet Solis-Gabriel performed the experiments, analyzed the data, prepared figures and/or tables, and reviewed drafts of the paper.
- Wendy Mendoza-Arroyo performed the experiments, analyzed the data, prepared figures and/or tables, and reviewed drafts of the paper.
- Karina Boege conceived and designed the experiments, performed the experiments, analyzed the data, contributed reagents/materials/analysis tools, wrote the paper, and reviewed drafts of the paper.
- Ek del-Val conceived and designed the experiments, performed the experiments, analyzed the data, contributed reagents/materials/analysis tools, wrote the paper, prepared figures and/or tables, and reviewed drafts of the paper.

## Data Availability

The raw data has been supplied as Supplemental Dataset Files.

## Supplemental Information

Supplemental information for this article can be found online at http://dx.doi.org/10.7717/peerj.3344#supplemental-information.

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
