# Peer review of "Restoring lepidopteran diversity in a tropical dry forest: relative importance of restoration treatment, tree identity and predator pressure"

_PeerJ, doi:10.7717/peerj.3344_

## Round 0.1 · original submission · Minor Revisions

· Academic Editor

Minor Revisions

The reviewers have recommended various revisions, mainly to clarify the text and results. Reviewer #2 has uploaded comments as an attachment. In addition, I have the following comments/corrections:

L 19 Identify abbreviation in parentheses after first use “(TDF)”
L 113 “sites were” should say “sites where”
L 120 Sampling was started 3 years after initial planting. Please clarify whether the treatments (grass cutting, continued maintenance of or replacement of degraded plastic) were still ongoing during the sampling period. If the treatments (every 3 months) were no longer being applied this could make it difficult to detect treatment effects (they would best be called legacy treatment effects if the treatments were no longer ongoing by the time you sampled the sites).
L 149 At what rate did the clay naturally fall off the leaves (not due to predation) due to failure of glue?Or, what was the % of the ignored clay models in the “disappeared” category (line 154).
L 216-219 – The evidence presented here is highly circumstantial. Readers’ confidence would be bolstered if you could describe a couple actual observations of attacks on the clay by birds and invertebrates. For example, I find it hard to believe that a parasitoid would oviposit on clay, since chemical signals from a live caterpillar are important elicitors. Likewise it seems unlikely that ants would attack clay having mistaken clay for a caterpillar. To demonstrate that this can indeed happen, please describe some of your observations. In the absence of any actual evidence that marks were caused by predators, these findings seem highly suspect.
L 270-272 These statements need justification in light of the comments above. If appears that you actually did not “document” any predation. It is only inferred from circumstantial evidence and wording should be softened accordingly, both here and in the abstract.

Figure 2 Caption – this figure does not show different restoration treatments. The caption should say “averaged across all restoration treatments”.
Figure 3 – legend is needed for colors (why are there different colors?)
Figure 4b use consistent placement of error bars – on figure 4a they are both above and below each bar. It would be better not to use solid black bars since they obscure the error bars (also on fig 3), or just show error bars above each bar on all of the graphs.

Reviewer 1 ·

Basic reporting

Overall the English is good, but there are a number of minor issues that need some attention.

Line 59: typo – “They use” needs to be changed to “They used”
Line 78: wrong word – “imply” to “lead to” (or something like)
Line 103: typo – “block” to “blocked”
Line 104: missing word “and A control”
Line 112: Can you ensure mycorrhizal colonization and did you check? If no to either of these, use the word “facilitate” rather than “ensure”.
Line 119: “In order to assess immature Lepidopteran diversity” change to “In order to assess Lepidoptera larval diversity”
Line 122: You don’t need to repeat the treatments here, they are clear by this stage.
Line 140: Why did you only use two of the treatments to look at larval predation? If it was due to time constraints, then say this.
Line 153: Change “Disappeared” to “missing”. Also why did they disappear as they can’t walk off the plant?
Line 157: Change “Lepidopteran diversity (richness and abundance)” to “Lepidopteran richness and abundance” as diversity isn’t richness & abundance per se.
166: lme – put in full (it’s a statistical shorthand I think?)
Line 212: How much more predated were the green caterpillar models than the brown ones? Ie give some means as well as the statistics.
Line 218: I’m very doubtful that parasitoids would drill small holes in clay caterpillars. They will use a suite of cues to recognize a caterpillar (scent, texture etc) and I don’t think that they would be duped by clay ones as easily as predators. Unless you actually saw them drilling the holes, remove this from the text.

Experimental design

Overall the design is very good. I have three queries that need addressing though:

1) There is no underlying hypothesis for why they used clay caterpillars of two different colours.

2) Does plastic mulching affect water flow through to the roots – probably not given there is no difference to the control but it would be good to clarify this.

3) There needs to be a figure showing the layout of the experiment (in a in a supplementary information section if space is short) in order to check that there are no environmental confounding effects in their design, ie that treatments are randomly placed within the overall field site and there are no spatial confounding effects. Confounding effects are unlikely given the lack of significance for the treatments, but it’s really important to get this part of the design right and they should make the spatial arrangement of plots clear to the reader.

Validity of the findings

The research has a robust experimental design which is fit for purpose (thought it would be good to check for spatial confounding), it’s well written and interesting to read, it’s in a topical area (restoration ecology) and provides an unusual angle on it too. It’s also pragmatic in the advice that it gives – going for the most economical treatment given the restoration treatments don’t make much difference (rather it’s a tree species effect that underlies the pattern of variation).

Additional comments

I enjoyed reading this paper which compares three different methods (mulching, cutting grass and control) of restoring tropical dry forest using a robust experimental design. There are many studies which use this sort of design to look at plant growth and survivorship, but this one does something much more interesting it asks how the treatments affect caterpillar establishment and caterpillar predation. Thus the authors are asking about the restoration of the second trophic level, and the effect of the third and this gives the work a rather novel perspective.

The authors have missed an easy win in their introduction (line 51) in that herbivores are also food for the next trophic level and so have an added importance above and beyond restoring Lepidopteran diversity and abundance as they are needed to restore the insectivore community.

Reviewer 2 ·

Basic reporting

The article has some spelling and grammar mistakes, but in general is easy to read and has a good sentence structure; nevertheless I am not an English spoken and I am not the indicated person to give specific recommendations.

It has a good introduction, synthetic but informative and with appropriate references, but there are some literature references not cited in the text and some references are incomplete or with mistakes, for example:

Tvardikova, K. and V. Novotny. 2012. Predation on exposed and leaf-rolling artificial caterpillars in tropical forest of Papua New Guinea. Journal of Tropical Ecology 00:1-11.

( it must say: 28: 331-341)

González-Tokman, D. M., V. L. Barradas, K. Boege, C. Domínguez, E. del-Val, A. Mendoza, E. Saucedo, and C. Martínez-Garza. Mulching maximizes growth rates but not survival of 11 tree species in restoration plantings in a Mexican dry forest.

I couldn’t find this reference (Year??, journal??)


The sections of the article are accord with a scientific paper and follow the recommendations of the journal, but they need to adjust the in-text citations.


The figures have a good resolution but need changes because some are incomplete, with wrong names or labels. I give the observations in the correction document.
The raw data are given in the supplementary document.

They attend all the investigation questions with pertinent analysis following the described methodology.

Experimental design

This research is part of a global project in forest restoration, it is relevant and has a background related to first aspects of the well-formulated questions.

All the techniques used are valid and conducted in an appropriate manner.

They describe the methods in a general way and recommend a reference if somebody needs more information, the problem is that the citation in the text and the reference is not correct and I couldn´t find in any catalogue in the web the paper that they suggest (see the basic report).

Validity of the findings

In a broad sense, the results are interesting and contribute to answering the established questions. But the presentation in the text has inconsistencies because they have inaccuracies in the figures; also I corroborate some results with the raw data and detected some mistakes. I made some suggestions in the pdf.

It is very important that they review carefully all the figures and fixed the related number with the text.

The discussion is related with the results and supported with references.

Additional comments

Please attend the recommendations included in the amendments document.

Annotated reviews are not available for download in order to protect the identity of reviewers who chose to remain anonymous.

---

## Round 0.2 · Minor Revisions

· Academic Editor

Minor Revisions

I agree that the manuscript is improved. I made a number of English edits on the Word manuscript using Track changes. Unfortunately, the system only allows me to upload PDF format, so you should find the PDF with corrections on the online system. If you prefer Word, let me know and I can send it directly to you by email. I also have a comment about page 5 line 13 -- Your letter says the treatment was only applied once. Please clarify on p 5 line 13 how many times this cutting treatment was applied.

---

## Round 0.3 · accepted · Accept

· Academic Editor

Accept

Thanks for following up with the requested corrections. I am recommending your manuscript for publication in PeerJ.